# In Search of Digital Dopamine: How Apps Can Motivate Depressed Patients, a Review and Conceptual Analysis

**DOI:** 10.3390/brainsci11111454

**Published:** 2021-11-01

**Authors:** Stephane Mouchabac, Redwan Maatoug, Ismael Conejero, Vladimir Adrien, Olivier Bonnot, Bruno Millet, Florian Ferreri, Alexis Bourla

**Affiliations:** 1Department of Psychiatry, Hôpital Saint-Antoine, AP-HP, Sorbonne Université, 75012 Paris, France; vladimir.adrien@aphp.fr (V.A.); florian.ferreri@aphp.fr (F.F.); alexis.bourla@aphp.fr (A.B.); 2iCRIN (Infrastructure for Clinical Research in Neurosciences), Brain Institute (ICM), Sorbonne Université, INSERM, CNRS, 75013 Paris, France; redwan.maatoug@aphp.fr (R.M.); b.millet@aphp.fr (B.M.); 3Sorbonne Université, AP-HP, Service de Psychiatrie Adulte de la Pitié-Salpêtrière, Institut du Cerveau, ICM, 75013 Paris, France; 4Department of Psychiatry, CHU Nîmes, University of Montpellier, 30090 Nîmes, France; ismael.conejero@gmail.com; 5Inserm, Unit 1061 “Neuropsychiatry: Epidemiological and Clinical Research”, 34000 Montpellier, France; 6CHU de Nantes, Department of Child and Adolescent Psychiatry, 44093 Nantes, France; olivier.bonnot@chu-nantes.fr; 7Pays de la Loire Psychology Laboratory, EA 4638, 44000 Nantes, France; 8Jeanne d’Arc Hospital, INICEA Korian, 94160 Saint-Mandé, France

**Keywords:** dopamine, digital phenotype, EMI, EMA, depression

## Abstract

Introduction: Depression is highly prevalent and causes considerable suffering and disease burden despite the existence of wide-ranging treatment options. Momentary assessment is a promising tool in the management of psychiatric disorders, and particularly depression. It allows for a real-time evaluation of symptoms and an earlier detection of relapse or treatment efficacy. Treating the motivational and hedonic aspects of depression is a key target reported in the literature, but it is time-consuming in terms of human resources. Digital Applications offer a major opportunity to indirectly regulate impaired motivational circuits through dopaminergic pathways. Objective: The main objective of this review was twofold: (1) propose a conceptual and critical review of the literature regarding the theoretical and technical principles of digital applications focused on motivation in depression, activating dopamine, and (2) suggest recommendations on the relevance of using these tools and their potential place in the treatment of depression. Material and Methods: A search for words related to “dopamine”, “depression”, “smartphone apps”, “digital phenotype” has been conducted on PubMed. Results: Ecological momentary interventions (EMIs) differ from traditional treatments by providing relevant, useful intervention strategies in the context of people’s daily lives. EMIs triggered by ecological momentary assessment (EMA) are called “Smart-EMI”. Smart-EMIs can mimic the “dopamine reward system” if the intervention is tailored for motivation or hedonic enhancement, and it has been shown that a simple reward (such as a digital badge) can increase motivation. Discussion: The various studies presented support the potential interest of digital health in effectively motivating depressed patients to adopt therapeutic activation behaviors. Finding effective ways to integrate EMIs with human-provided therapeutic support may ultimately yield the most efficient and effective intervention method. This approach could be a helpful tool to increase adherence and motivation. Conclusion: Smartphone apps can motivate depressed patients by enhancing dopamine, offering the opportunity to enhance motivation and behavioral changes, although longer term studies are still needed.

## 1. Introduction

### 1.1. Motivation Deficit in Depression: A Neglected Dimension

Depressive disorder manifests in all three components of the mind. The affects, which translate to the emotional interpretation of perceptions and representations of the world, are inherently negative in depression. Cognition refers to the processes that allow one to have knowledge and understanding of the world. Finally, the conation, which is crucially important and which forms the junction between the two preceding components, refers to goal-directed behavior, intentionally based on motivational factors.

From a theoretical point of view, appropriate decision making relies on the ability to evaluate the expected costs and benefits of an action. The net value of an option therefore corresponds to the benefits of the action (when the reward is greater than the constraints) from which we will subtract the costs associated with it (the nature, quantity and duration of the efforts to be provided). If more than one choice is possible, then the one with the highest positive net worth will be preferred. Motivation determines the triggering of goal-directed actions and the appropriate intensity, and above all it ensures the prolongation until the end or the interruption. In this perspective, motivation, which will allow oneself to make these choices, can have several dimensions involving distinct mechanisms: a first, which corresponds to the identity of the goal and therefore its content (“what”); a second, which refers to the value of the goal (“how much”); and the last one which corresponds to the process allowing to adjust the goal-directed behavior (“how”). A simple action, such as entering a restaurant, depends on several parameters: the level of hunger, our culinary preferences and the person whom we will eventually meet. However, all of these parameters can be linked to an expected pleasure. Psychiatrist D. Klein noted that depressed patients with anhedonia seemed to be able to subjectively experience pleasure, but only for readily available rewards. Furthermore, they complained of not feeling any urge to obtain them [1].

Anhedonia refers to the loss of the ability to feel positive emotions, a form of insensitivity to pleasure found in depression and which contrasts with the subject’s previous state. In a study of 1570 depressed subjects, we showed that reduction of anhedonia was the strongest predictor of improvement in psychosocial functioning. In addition, it appeared that the reduction of anhedonia over time was a mediation factor between the improvement in depressive symptoms and the improvement in social functioning [2].

Furthermore, we showed an overlap with anhedonia and other symptoms such as apathy, which belongs to the motivated behavior disorders and results in a loss of the ability to feel emotions (positive or negative) or initiate spontaneous behaviors. Clinically, there is a loss of interest and pleasure in daily activities in patients living with apathy. Abulia, very similar to apathy, refers to a lack of will or initiative, and can be seen as a disorder of diminished motivation. The condition was originally considered to be a disorder of the will, and people suffering from abulia are unable to act or make decisions independently; however unlike apathy, abulic patients remain sensitive to the reward if they complete a task.

We therefore understand that the motivational dimension is an important target in the treatment of depression and that the specific treatment of conative disorders, including anhedonia, abulia or apathy is central. Likewise, contemporary approaches propose to distinguish several components in anhedonia [3].

Each is based on distinct neurobiological circuits modulated in part by dopamine:

Motivational anhedonia, which corresponds to a decrease in the orientation of action according to the anticipated response to a stimulus (“wanting”). The representation of an incentive potential reward is altered and the amount of work that can be done to achieve it is lower. Consumption anhedonia, which corresponds to a decrease in the perceived reward for pleasure during the action (“liking”). More recently, a learning component has been added which makes it possible to use past hedonic experiences to optimize future predictions (“learning”), so this is a learning anhedonia.

In summary, when engaged in a task, the depressed patient will have a greater sensitivity to effort and less sensitivity to reward [4]. This will affect future engagement in similar tasks.

These three clinical dimensions of depression interact with each other: negative thoughts can generate negative emotions; negative emotions then prevent us from acting. So, lack of action sustains negative cognitions.

While we cannot reduce each of the clinical dimensions of depression to a single neurotransmitter, it is clear that dopamine deficiency plays a critical role in altered motivated behaviors [5,6]. Indeed, the dopaminergic system promotes the development of motivation and the triggering of adaptive behaviors [7]. This system is activated by pleasant or aversive stimuli and therefore helps to adapt to changes, thus promoting survival. The attribution of an emotional valence to the events is fundamental and will be supplemented via the reward system [8]. Finally, at the cognitive level, dopaminergic neurons are involved in attentional processes and the learning of new situations [9].

### 1.2. A Specific Approach Is Needed to Treat Conative Disorders

Pharmacological treatments act only partially on this dimension, and direct stimulation of the dopaminergic pathways may present certain risks (dependence, triggering of pathological gambling, purchasing or hypersexuality). The modes of action of some newer antidepressants that indirectly activate this pathway are therefore interesting. However, we know that given the clinical heterogeneity of depression, more specific approaches are necessary. Furthermore, when emotional disturbances are major, classically, a tendency to “anesthetize” them is observed. This result in avoidance behaviors (procrastination, ruminations, avoidance justified by certain symptoms including fear, social isolation): TRAPs “Trigger Response Avoidance Pattern” must therefore be replaced by TRACs “Trigger Response Alternate Coping Response”. For this purpose, we have to act on improving the perception of inciters or reinforcers and optimize their mental representation during the motivational stage and promote them. We must also help the individual to become more aware of his positive actions in order to optimize his reinforcement processes, because the depressed patient overly activates the punishment system.

In this perspective, the activation therapies are well suited: they are brief, structured approaches that seek to increase the engagement of the depressed subject [10]. He must be attentive to his emotions, choose suitable alternative behaviors and test them. After having observed their positive results, he will have to integrate the profitable changes and especially not to be discouraged in the event of partial failure. Among other things, the effect produced by the actions results in an improvement in mood, the ability to think and solve problems. Because this process does not happen all at once and requires daily work, repeated evaluations and measurements, and above all, substantial therapeutic support, are required. They are more ecological, that is to say, not constrained by an external influence, and are supported by the theory of self-determination. Motivation in this model depends on the fulfillment of three universal basic needs: autonomy (having the feeling of functioning independently reinforces the commitment of individuals), the competence which corresponds to the feeling of efficiency and control over the environment (obtaining positive feedback from the environment following a task increases motivation) and finally relatedness (strength of the reciprocal link between individuals). The first two needs are strongly linked to the concept of intrinsic motivation, defined by an activity carried out by the subject according to his value system (expected pleasure). It is opposed to extrinsic motivation, which corresponds to the pursuit of an activity to achieve an external goal, under the constraint of environmental factors.

### 1.3. Digital Dopamine to the Rescue

For thousands of years that have shaped our evolution, hominids have therefore acquired cognitive and conative skills to adapt to the environment. They were optimized to solve the problems they encountered during this period which concerned their survival. They are called “heuristics”. Their goal is to find effective, quick solutions while saving mental energy. In the original hunter–gatherer environment, these cognitive tools improved our analytical and reaction skills, but in our modern world with its complexity, they become unconscious relics that sometimes lead us to make irrational decisions, which are now called “cognitive biases”. This relies partially on the powerful phylogenetic circuits of reward and punishment, which will be “modulated” [11].

Some tech industries use these neuroeconomic attributes to capture the subject’s attention and keep it in concentrated on monetizable activities. This is called neuromarketing, and digital marketing is one of the applications: numerous apps or WEB interfaces are optimized for this objective. Each “like” or badge causes a shot of dopamine and maintains the individual in a consumer role. Social interactions in digital networks are levers to stimulate dopamine (a kind of “numeric sugar”).

A recent meta-analysis shows that the prevalence of addictions to social networks ranged from 5% to 25% depending on the type of classification and the level of addiction used: 5% for strict monothetic classifications, 13% for severe levels and 25% for a moderate level [12]. Studies show that the impulsive amygdala–striatal system is more reactive, and that the reflective–inhibitory prefrontal brain system is not hypoactive, which is different from other addiction models [13]. If this is the “dark side” of digital dopamine, a form of brain hacking exploiting people’s psychological vulnerabilities, can we now transform it and use these technologies for therapeutic purposes and for changing depressed people’s behavior by enhancing digital dopamine production?

### 1.4. The Role of Digital Technologies: Toward the “Light Side”

E-health includes all of the health-related activities that are based on information and communication technologies. However, we will focus our discussion on mobile health, as in those that use mobile phones, connected objects and dedicated software, and enlighten practitioners on the various aspects of the development of these tools. An increasing number of new tools are emerging for the assessment and treatment of psychiatric disorders, and the promises of e-health are numerous; they are disrupting our practices without replacing them and offering tremendous opportunities [14,15,16].

So, the stakes are high, with a market estimated at more than 10 billion connected objects in 2020, in 2013 80% were related to health and more than 100,000 mobile health applications are available today. Finally, “health—well-being” downloads on the “stores” are in the region of hundreds of millions. This is a growing market and of course, the advertisements for apps are very attractive and sometimes have a very effective advertising dynamics. While the intention may be good, the reliability of these tools is sometimes highly questionable [17]. Therefore, APA reports that there is a gap between positive statements from developers about the quality and goals of their apps and the level of scientific evidence they can demonstrate.

The development of the concept of a “digital phenotype” (DP) may make it possible to improve the diagnosis of mental illnesses and optimize curative or preventive therapeutic actions. Introduced by Sachin Jain in 2015 [18], the paper argues that collecting real-time data on human behavior and digital markers of their functioning could characterize a “digital signature of a pathology” [13]. This approach is based on the collection of so-called “passive” data, since in most cases it is no longer necessary to intervene during the data collection phase, which takes place automatically without the individual being aware of it (GPS or accelerometer, physiological information from peripheral sensors [19], analysis of the nature of communications, the emotional content of SMS, voice and language analysis and the time spent on screen) [20].

On the other hand, DP involves the analysis of “active” data, which involves the patient in data collection. The use of the Momentary Ecological Assessment (EMA) is the most common and practical method [21]. The patient can be interviewed regularly to gather clinical information, even automatically at scheduled times of the day or in response to changes in “passive” data. The valence of emotions, the level of energy and motivation or the presence of symptoms with their perceived intensity (ruminations, suicidal thoughts) can be analyzed and are well suitable. The cross-use of this information data then make it possible to propose personalized interventions [22], or even to predict the occurrence of clinical events when the use of artificial intelligence is introduced.

In total, 70% of people suffering from mental disorders do not receive adequate psychological treatment or reach complete clinical remission [23], and in order to offer treatments to patients between therapy sessions, Ecological Momentary Interventions (EMI) are ideal tools [24]. We can suppose that EMI are also an effective way to apply the principles of activation therapies. They therefore refer to interventions offered in real time, in natural environments and above all, accompanying daily life. These are therefore extensions of psychotherapies that will engage the patient in beneficial activities and promote the strengthening of individual skills to increase, among others, the hedonic experience as we have described previously. Smartphones are particularly suited to this type of process, and coupled with machine learning algorithms, it would be possible to numerically define an individual level of activation (“e-dopamine” level) and to propose specific interventions adjusted to these levels.

Innovations in the field of digital health have the potential to disrupt current practices, and healthcare practitioners have to be informed about the opportunities offered by these new technologies. Even if we know that the development of these tools remains heterogeneous, we conduct a review to evaluate the usefulness and efficiency of new technologies using Ecology Momentary Interventions for motivation in the field of depression.

## 2. Materials and Methods

Several Smart-EMI apps which have a goal to reduce depression have been created and evaluated. In this section, we conduct a narrative review using the MeSH terms and keywords [ecological momentary intervention], [digital], [smartphone], [app], [computer], [computerized], [software], [serious game], [web-based], [internet], [social network], [Facebook], AND [motivation], [reward], [pleasure], [dopamine], [depression]. We searched the PubMed database for articles (including conceptual ones) in the field of smart-EMI, summarizing the evidence specifically for goal-oriented behavior enhancement, motivational increase, reward system triggering or hedonic improvement. AB, RM and SM screened 1366 titles, 193 abstracts were included, and 29 articles were then analyzed.

Inclusion criteria were articles that integrate data collection (EMA) and intervention (EMI) in order to enhance motivation. These could be:-Both an EMA and an EMI component that are not connected to each other.-EMA triggered by another EMA component.-Fully or partially automated intervention based on the EMA components allowing tailored interventions in real time.

Exclusion criteria: articles without motivation or mood or activity in the evaluated outcomes and those for which no data collection was used (for example, articles with only one intervention and no EMA).

## 3. Results

### 3.1. Definitions

EMI differ from traditional treatments (internet treatment, iCBT, etc.) by providing relevant, useful intervention strategies in the context of people’s daily lives. EMI can be triggered by EMA (a process called “Smart-EMI” while “Smart-EMA” refers to EMA triggered by another EMA component), see Table 1. Smart-EMI therefore consists of a combination of interventions and decision rules that specify when and why those interventions will be deployed. That procedure can mimic the “dopamine reward system” if the intervention is tailored for motivation or hedonic enhancement, and it has been shown that a simple reward (such as a digital badge) can increase motivation [25]. Other EMIs mimic psychotherapy, with a motivational dimension such as acceptance and commitment therapy [26], interpersonal therapy [27] or behavioral activation [28]. For review, see [29,30].

Real life training (“in situ”) appears highly relevant for learning or to regain healthy behaviors, since depressed people regularly switch from goal-directed behavior to habits [30], and EMIs are appropriate tools since they can be implemented in real-life settings which may provide a more effective way to stimulate dopamine secretion. Schueller et al. [42] provide a model called the “Behavioral Intervention Technology (BIT) model”, first seen in Mohr et al. [43], referring to a specific single interaction between a user and an element of technology, for example a single push notification is a BIT intervention while a BIT treatment refers to the sum of these interactions that unfold over time. In order to activate the reward system, a decision rule (referring to the workflow characteristics that define when a specific intervention will be deployed) has to be made with EMA-based triggers (active date), time-based rules or passive data gathering (e.g., GPS, phone activity, screen state, smart watches that may provide heart rate monitor, accelerometer, etc.). Several complexity levels can be determined depending on the relationship between the user and the intervention. For example, a high complexity level is represented by “Just-In-Time Adaptive Interventions” (JITAI) that are becoming more prevalent for mental health apps, with intervention automatically triggered by a contextualized situation. For depressed patients with apathy, a Smart-EMI or JITAI intervention could be a motivational message sent to the patient if the smartphone detects a long period of lying down. A low complexity level is represented by interventions tailored for a single purpose:-Engagement in pleasurable activities [44];-Increasing positive emotions;-Motivate the patient;-Encourage engagement in practicing or using previously learned skills.

### 3.2. Applications

Soares Teles et al. [36] describe a Smart-EMI app with two components: **Situation Manager** (SituMan), a mobile system that provides situation awareness to the mobile applications (such as “working”, “studying”, “socializing”, etc.) using contextual information obtained from sensors embedded in mobile devices and specified by the user. This component is the EMA module enriched by passive data and it allows patients to express interest for depression treatment, to identify when stressful situations happen, and to trigger the second component. **MoodBuster**, the Smart-EMI component, requests mental status self-assessments performed in preset situations by asking questions about the patient’s state at various moments of the day. It presents accumulated answers in a graph which allows feedback to the patient on their own behavior and symptoms. This feedback can be used to activate the reward system.

Everitt et al. [31] have performed a naturalistic, 3-week, randomized, waitlist-controlled trial on 235 participants with three different interventions:-**MoodTracker**, an EMA only app-**ImproveYourMood**, an EMA and EMI app-**ImproveYourMood+**, an EMA and EMI app + prompts

They find that the effect of EMA only was logically similar to the waitlist condition, and that both ImproveYourMood and ImproveYourMood+ improve depressive symptoms. They conclude that even microinterventions can be an effective way to reduce depressive symptoms up to 2 months later.

**CBT2go**, developed by Depp et al. [32] is an EMA + EMI mobile app designed to provide real-time thought-challenging intervention, individualized to specific symptoms, for people with serious mental illnesses (schizophrenia and bipolar disorder). Several questionnaires about maladaptive beliefs, socialization, and medication adherence can trigger an intervention offering potential alternative or adaptive belief, personalized to the individual in the in-person session. This intervention is backed by follow-up telephone contacts. An RCT on 255 individuals showed that a single intervention augmented by CBT2go was feasible, and was associated with small yet sustained effects on global psychopathology.

**Mobilyze!** [28] is a Smart-EMI, internet-based intervention, combining an EMI component with online behavioral skills training and human email support. The EMA component provides the mood state of the patient five times a day and triggers tailored feedback in the form of a message being sent reinforcing improvement or suggesting a website tool in the case of deterioration (i.e., didactic lessons on behavioral activation). A single-arm pilot study was performed with the aim to help patients gather insight into their daily activities and behavior, and to engage more often in some of them. Depressive symptoms decreased significantly over time.

**PsyMate** [33,45] is an EMA + EMI intervention that consists of standardized feedback on personalized patterns of positive affect. To study its effectiveness, 102 depressed outpatients were randomized into three arms:EMA+ EMI feedbackEMA (called “ESM” in this study) only.Treatment As Usual (TAU).

EMA + EMI and EMA only induced a clinically relevant and statistically significant decrease in depressive symptoms with an increase in talking and a decrease in doing nothing/resting or being alone. In these studies, the feedback was provided in a weekly face-to-face contact session with the researcher, making it an intensive treatment that could have impacted the results.

Help4Mood [34] is an EMA + EMI intervention. The EMA component is a self-monitoring web platform collecting data on symptoms, activities, thoughts, and passive data (actimetry sensor and acoustic analysis of speech). The EMI is a digitalized CBT approach mediated by a virtual conversational agent helping patients to reflect on the emotional and cognitive patterns related to depression. A multicentric pilot RCT study was performed, and the results showed moderate acceptability with a mild retention level. Only individuals who used Help4Mood regularly reached significant reduction of their symptoms.

**Therap-i** [37] has an EMA component supplemented with personalized items to cover core elements tailored for each individual (case concept). In this intervention, patients and their clinicians collaborate together with the researcher in order to personalize data collection and EMA-derived feedback. The feedback includes graphs on daily fluctuations of scores, and their associations with contextual information. Patients fill out their personalized diary five times a day for eight weeks (the system is programmed to send to the patient’s smartphone a notification with a link to the online questionnaires). Patients receive personalized feedback during a regular consultation with their clinician and the researcher after two, four, and eight weeks of monitoring. Data collection is ongoing.

**LiveWell** [35] is another low complexity EMA + EMI user-centered smartphone app used primarily for bipolar disorder patients. The EMA component is Daily Check-in (sleep, routine, wellness, medication, expectations, social interactions, other personal goals) associated with “Daily Review” bar graphs that indicate the percentage bars corresponding to goal achievements over the past week. The EMI component consists of several recommendations (“lifestyle plan”), such “pay attention to sugar” “exercise at a consistent time” “go to work with a positive attitude” “limit caffeine”, etc., or contacting the provider if the EMA component detects that assistance is needed. Email alerts to coaches and coaching scripts were developed to help coaches contact the user and a psychiatrist if needed. At the end of the pilot study, users reported that the EMA made them more aware of early warning signs (such as sleep time) and symptoms. Users did not consistently follow up with the EMI recommendations to contact providers, so the human support roles for the technology were expanded beyond app-use support to include support for self-management and clinical care communication.

**PRIME** (Personalized Real-time Intervention for Motivational Enhancement) [38] consists of a mobile intervention aiming at improving motivational impairments early in schizophrenia. It includes an EMA component (goal and achievement tracking), a smart-EMI component (i.e., EMA triggering a display of brief challenges such as “invite a family member to do something fun with you”), human intervention (with CBT-based coaching by master-level clinicians) and a peer community (supportive online environment). The intervention was designed to target the motivational system using social reinforcement to engage and sustain goal-directed behavior (requiring successful engagement in the various component processes of reward processing). Schlosser et al. designed a 12-week, fully remote, randomized control trial on 43 people with recent-onset schizophrenia. A total of 22 people were randomized to PRIME, and 21 to TAU. Motivational assessment was performed at baseline, at the end, and 3 months after the end, and showed that the PRIME condition showed significantly greater improvements in self-reported depression, defeatist beliefs, self-efficacy, and a trend towards motivation- and pleasure-related symptoms post-trial, persistent at the 3-month follow-up.

**MASS** (Motivation and Skills Support) [39] is a mobile smart-EMI program also tailored to patients with schizophrenia, aiming to specifically increase their motivation and goal-oriented behaviors. The EMA component consists of push notifications three times per day, directing users towards app content focused on their identified social goals and steps, and a survey or questionnaire to assess their motivation (i.e., “How motivated are you to work on this step?”), and custom EMI feedback depending on their response to the EMA (i.e., “You made a moderate amount of progress” or an encouragement “Do not be too hard on yourself”). Even if the MASS app was designed for schizophrenic patients, it shows promise as a scalable digital intervention tool for motivation enhancement and support in the pursuit of social goals.

**ICanSTEP** [40] consists of an EMA module (activity tracker) and a Smart-EMI component (daily text messages personalized to their activity level) to improve and promote physical activity through self-activation in cancer survivors. A nonrandomized prospective intervention trial was performed on 30 patients, and while the primary endpoint was the 6 min walk test at 3 months, the secondary result was a significant improvement in the Beck Depression Inventory. These results highlight that a smart-EMI intervention aimed at promoting physical activity has a motivating effect.

**Hiroshima Health Note [41]** is a web app with an EMA component based on sensors, recording physiological signs (built-in features to monitor heart rate, daily steps, blood pressure, body weight, etc.) and an EMI component sending autogenerated messages that remind the user to measure and record physiological signs, and messages that encourage participants to review their own data and to continue with behavioral changes. This program was not developed for depressive disorders since it was designed for increasing adherence to lifestyle changes and improving indicators of metabolic disturbances among Japanese civil servants, but it has demonstrated that it is possible to use EMI feedback based on physiological signs alone in order to motivate participants to display a behavioral change. Although, it is possible that those changes do not last after the observation period. Other studies on how to improve physical activity and motivation for behavioral changes have been performed with positive results [46,47,48,49].

Some authors almost compare Smart-EMI to a new form of biofeedback [50]. Biofeedback is a painless, noninvasive procedure that consists of capturing biometric data (i.e., EKG, EEG, skin conductance, temperature, etc.) and feeding them back to the patient instantaneously, with the objective of modeling the patient’s brain activity in real time as an image (video-game type) or a sound in order to use CBT techniques (or relaxation) to promote brain activity corresponding to the therapeutic target [14]. Smart-EMI is therefore a form of biofeedback, where the biometric data is replaced with EMA (active or passive data gathering), and the EMI component (regularly using CBT techniques) promotes behavioral changes or motivation enhancements.

For a synthetic view see Table 2.

## 4. Discussion

The various studies presented support the potential contribution of digital health to effectively motivating depressed patients to adopt therapeutic activation behaviors. Although longer term studies are needed.

Finding effective ways to integrate EMIs with human-provided therapeutic support may ultimately yield the most efficient and effective intervention method. This contact could be a helpful tool to increase adherence and motivation, which in turn could result in a stronger effect [42]. This is consistent with Versluis et al. [30] who identified 17 studies on the effects of EMI on depressive symptoms and found that EMIs had a small-to-medium effect on depressive symptoms, and they highlight the important impact of additional human support. Another systematic review on that subject [29] concludes on the limitation of EMAs and EMIs, since these approaches are still time-consuming for now and might be perceived as invasive by patients since they are required to complete multiple assessments throughout the day, and protocols often last weeks. Another important concern is about data security and the fact that people might not be willing to share personal information. Bos et al. [51] therefore recommended that EMAs should be implemented by an interdisciplinary team (patients, clinicians, researchers, technology specialists) since although clinicians and patients recognize that EMA + EMI could be relevant for increasing self-management, they warned against high assessment burden and potential symptom worsening [21].

We could be criticized for a form of paradox in this approach, namely on how to motivate a disabled person in this area between two consultations often distant in time.

For some authors, engagement, which corresponds to the act by which one commits to accomplish something, is not a static process and relies on several stages. First, there is an engagement point; second, there is an engagement period; third, a disengagement time; and finally, fourth, a possible re-engagement period. It is often observed that in the event of a decrease in behavioral symptoms, engagement decreases.

It is in fact reported that the use of health applications follows a “law of attrition”: optimal use is observed only over a short period, then the rate decreases. Only 4% of behavioral health app users continue to use the app after 15 days [52]. Reasons for this loss of participants have been described by Eysenbach [53]. However, he argues that this rate decreases if the user perceives that this innovation is superior to the idea it replaces. The explanation of the potential advantage must therefore be provided by the prescriber.

Recently, authors have proposed a four-dimensional scale that measures the subject’s level of engagement in a technology [54]: the flow or level of focused attention—which partly reveals the cognitive load required for the tasks; the quality of usability (i.e., intention to use and ease of use) depending in the part on the positive experience; the aesthetic appeal, which has an emotional component too; and the perception of the rewards obtained. This last dimension of reward is probably the most interesting for our purpose because it supports the hedonistic aspect of the experience and must be developed, and it must also take into account the motivational deficits specific to depression. Engagement is also conditioned by the nature of the experience or interactions with the apps. Additionally, “push factors” (reminders) and positive feedback are effective in keeping the subject engaged. In recent work evaluating the rate of use of an application aimed at parents of autistic children, we showed that the frequency of use of the feedback screen was correlated with a lower rate of attrition [55].

Thereby the cognitive, emotional, and behavioral components are often reported in user-engagement literature. Thus, the gamification approach is therefore an interesting option but still needs to be scientifically evaluated [56]. This corresponds to the use of gaming elements for nonplay purposes, and is based on the following fundamentals [57]: the significant objectives must correspond to the interests of the user; the apps must give users the power to act flexibly to achieve their goals; the personification is based on the exploitation of the individual characteristics of the patient (increasing levels of autonomy); as we have seen, the comments are important and highlight how the user’s actions play a role in their progress; and finally, there must be a strong “visibility” which quantitatively shows the progress and the current expectations. Gamification is underrepresented in depression apps. However, the application of this approach to mental health should not be just a transposition of methods aimed at gamers [58]. For example, when specifically adapted for cognitive training, gamification aims to increase engagement with boring tasks and increase long-term use, but some positive results should be taken with caution because of methodological problems [59]. Besides gamification, the importance of self-determination theory has also been discussed, which contributes to the improvement of intrinsic motivation by promoting autonomy, feelings of competence and relationships with other players.

Baumel et al. [49] suggest a set of conditions to build digital interventions that require little effort and are well-tailored for depressive disorder. We noted previously that in depressed individuals, the motivational component of hedonistic process is altered. So, making the task more attractive (i.e., more salient) and prioritizing it over other competitors is an effective solution. For this purpose, reminders could be proposed, offering immediate reinforcement with rewards and optimizing novelty (avoiding habituation) as needed. As sensitivity to the effort is increased, the tasks must be made as effortless as possible: they must be gradual and especially calibrated on the subject’s baseline. Finally, highlighting the link between the effort exerted and their engagement in the therapeutic process is fundamental: more explicit (narrative) interventions on the process carried out can help the learning step of hedonistic processes. An important condition is that all tasks occur in a real environment.

The strength of digital health tools lies in part in the possibility of initiating and supporting the change in an individual’s behavior over a more sustained period compared to intermittent visits to the medical office [60] As reported by Patoz et al. [61] numerous studies suggest that these applications are more appropriated to mild and moderate depression stages, considering that people suffering from severe depression would be unable to use an app. Regarding the acceptability, Litschitz et al. [62] have demonstrated that patients are interested in the app regardless of age and level of education, even if not familial with digital tools.

However, above all, they offer patients an additional path of empowerment. The goal in mental health is therefore for individuals to be able to exercise direct control over their decisions and events that could impact their daily lives, thus fostering the process that leads to internal power. Empowerment requires conditions for it to be put in place: the subject must participate in order to recognize and have their competence recognized, work on self-esteem and finally, be able to exercise critical awareness. Digital dopamine seems to be a good mediator to meet these conditions.

Although some tools have not been designed specifically for depression and do not take into account the different phases of the disease (prodromal symptoms, active phase, relapse, residual symptoms), the field of digital phenotypes seems promising. Moreover, it is complicated to draw too hasty of a conclusion, given the small number of randomized trials and the small sample sizes.

## 5. Conclusions

Smart-EMI (also sometimes called “EMI-feedback study”) uses aggregated EMA information to induce behavioral change. Integrating real-life assessments can create a completely new therapy, and as such, move beyond a mere extension of existing therapies into real-life contexts. These digitals tools, together with highly translational experimental models, are essential for the treatment of stress-related disorders such as depression and post-traumatic stress disorder [63]. This new digital—partially automated—therapy shows potential for motivation enhancement. Some authors [64] highlight the fact that all studies show broad feasibility and acceptability of EMA, EMI and Smart-EMI approaches in patients with depressive disorders or severe mental illnesses. They also call attention to the fact that data analyses report a significant efficiency of these procedures in mental health improvement, with a better level of evidence for efficacy concerning Smart-EMI (EMI integrated with real-life assessment using EMA) with tailored, personalized interventions, offering the opportunity to intervene with the day-to-day dynamics between the person and his/her environment in order to enhance motivation and behavioral changes.

## Figures and Tables

**Figure 1 brainsci-11-01454-f001:**
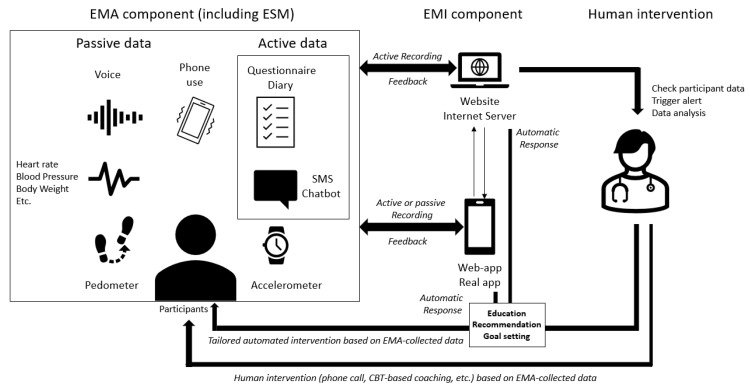
General design of Smart-EMI.

**Table 1 brainsci-11-01454-t001:** Definitions.

	Definition	References
EMA	Smartphone-based evaluation of day-to-day symptoms, in the habitual environment of the patient, with the possibility of withdrawing from recall biases since they evaluate themselves “Right then, not later; Right there, not elsewhere”. Experience Sampling Method (ESM) is a passive data-based EMA.	n.s.
EMI	Smartphone-based intervention involving the delivery of psychoeducation, advice, or recommendations about how to behave according to the patient’s immediate environment.	n.s.
EMA + EMI	Application that integrates both an EMA and an EMI component that are not connected to each other.	IYM [31] CBT2go [32] PsyMate [33] Help4Mood [34] LiveWell [35]
Smart-EMA	EMA triggered by another EMA component (for example a questionnaire that is triggered by a certain location, or by a cutoff on another questionnaire).	n.s.
Smart-EMI	Fully or partially automated intervetion based on the EMA components allowing tailored interventions in real time.	MoodBuster [36] Mobilyze! [28] Therap-I [37] PRIME [38] MASS [39] ICanStep [40] Hiroshima HN [41]

General design of SMART EMI is recalled in Figure 1.

**Table 2 brainsci-11-01454-t002:** Summary of applications and main results of studies.

App	Objectives	Population and Method	Result
SituMan	Accuracy of Situation Identification (SituMan component) EMA: Situation awarness Active data Passive data SituMan EMI: Feedback graph	12 healthy volunteers using SituMan for 7 days	Accuracy was 100% for three participants, >90% for five, >80% for three, >70% for one
MoodBuster [32]			
MoodTracker	Efficacy of microintervention content, just-in-time approaches, and the potential efficacy of symptom monitoring EMA: Mood active data EMI (IYM or IYM+ only): Breathing exercises Mindful body scan Gratitude exercise	Naturalistic trial on 235 healthy volunteers over 3 weeks, randomized into four groups: waitlist (control group), MoodTracker, IYM, IYM+	Participants in the IYM condition exhibited significantly greater improvements in depressive and anxietysymptoms (only at follow-up) and automatic negative thoughts (both postintervention and follow-up). EMI use resulted in immediate improvement in mood state, suggesting the resources had their intended effect in-the-moment
ImproveYourMood (IYM or IYM+) [27]			
CBT2go [28]	Assess the efficacy of interventions on Brief Psychiatric Rating Scale—expanded measuring psychopathologic symptoms (anxiety, depression, mania, delusions/hallucinations, unusual behavior, and negative symptoms) EMA: Maladaptive beliefs, socialization, and medication adherence Active data EMI: Psychoeducation about the topic and queried participants about their experience and current strategies for self-management	RCT on 255 participants diagnosed with schizophrenia, schizoaffective disorder, or bipolar I disorder randomized into three groups: Treatment as usual (*n* = 83), CBT2go (*n* = 77), Self-Monitoring (*n* = 69),	Participants who received interventions experienced greater improvement in global psychopathology than TAU. Community functioning improved more in the CBT2go vs. TAU condition
Mobilyze! [24]	Investigate the technical feasibility, functional reliability, and patient satisfaction EMA: Situation Symptom tracker Active data Passive data EMI: Behavioral activation approach	Eight adults with major depressive disorder in a single-arm pilot study for 8 weeks.	Intent-to-treat analyses revealed that depressive symptoms self-reported on the PHQ-9 decreased significantly over time
PsyMate [42]	Assess if ESM-derived personalized feedback can be used, in combination with standard antidepressant medication, as an effective add-on treatment for depressive symptoms EMA: Symptom tracker Active data (ESM) EMI: ESM-derived feedback (face-to-face contact)	Controlled trial on depressed patients randomly assigned to three arms: experimental (*n* = 33), pseudoexperimental (*n* = 36) or control group (*n* = 33) for 6 weeks.	The experimental group demonstrated a significantly greater weekly decline in depressive symptoms over the complete study period compared to the control group
Help4Mood [30]	Evaluate system use and acceptability, explore likely recruitment and retention rates and to obtain an estimate of potential treatment response EMA: Symptom tracker Active data Passive data EMI: Cognitive Behavioral Therapy	Multicentric RCT on 27 patients with MDD randomized to intervention + TAU or TAU alone for 4 weeks	ITT analysis showed a small difference in change of BDI-2 scores, but post hoc on-treatment analysis suggested that participants who used Help4Mood regularly experienced a median change in BDI-2 of -8 points
Therap-i [33]	Test the efficacy of the Therap-i module as a supportive tool in psychotherapeutic TAU in MDD patients EMA: Personalized items Active data EMI: EMA-derived feedback	Pragmatic RCT on 100 MDD patient randomized in the intervention group or TAU for 8 weeks	Data collection is ongoing
LiveWell [31]	Support the ongoing improvement and dissemination of technology-based mental health interventions. EMA: Wellness Plan Daily Check-in Active data EMI: Information on bipolar disorder self-management, Toolbox (skills practice), and Daily Review, lifestyle personalized plan for reducing risk	12 individuals with bipolar disorder participated in a field trial and an 8-week pilot study	Users reported that they were more aware of early warning signs and symptoms
PRIME [34]	Evaluate the efficacy of PRIME by assessing changes in components of motivated behavior using a modified version of the Trust Task EMA: Self-determined goals Active data EMI: EMA-triggered display of brief challenges, CBT, behavioral activation, mindfulness, and psychoeducation	RCT, 43 people with recent-onset schizophrenia spectrum disorders were randomized into the PRIME (*n* = 22) or TAU/waitlist (WL) (*n* = 21) during 12 weeks	Participants in the PRIME condition showed a greater increase from baseline to 12 weeks compared to WL
MASS [35]	Ongoing EMA: Social goals Steps Motivation Active data EMI: Custom feedback, encouragement and video clip made for improving social skills	Ongoing	Ongoing
ICanSTEP [36]	Evaluate whether wearable activity tracker with personalized text message feedback would increase physical activity. EMA: Activity Tracker Passive data EMI: Daily text messages personalized to their activity level	Pilot study on 30 patients with solid tumor cancers in a nonrandomized, prospective intervention trial lasting 3 months	39% of participants increased their steps taken by at least 20%, and 23% increased their 6 MW distance by 20%. At 3 months, there was a significant improvement in median BDI-II.
Hiroshima Health Note [37]	Evaluate whether an Information and Communication Technology (ICT) application motivated to increase adherence to lifestyle changes, and to improve indicators of metabolic disturbances EMA: Physiological signs Active data Passive data *EMI*: Tailored feedback Reminders encouraging participants to review their own data and to continue with behavioral changes	Nonrandomized, open-label, parallel-group study on 102 overweight or elevated glucose-concentration participants over 6 months. In total, 63 were allocated to intervention (ICT) and 39 to the control group	ICT group showed a significant decrease in male waist circumference, diastolic BP, and HbA1c and increase in HDL cholesterol

BDI-II: Beck Depression Inventory—Second Edition; BP: Blood Pressure; CBT: Cognitive Behavioral Therapy; EMA: Ecological Momentary Assessment; EMI: Ecological Momentary Intervention; HDL: High-Density Lipoprotein; MDD: Major Depressive Disorder; PHQ-9: Patient Health Questionnaire; RCT: Randomized Controlled Trial; S-EMI: Smart-EMI; TAU: Treatment as usual; WL: Waitlist.

## Data Availability

Not applicable.

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
