# Peer review of "In Search of Digital Dopamine: How Apps Can Motivate Depressed Patients, a Review and Conceptual Analysis"

_brainsci, 2021, doi:10.3390/brainsci11111454_

Round 1
Reviewer 1 Report
In their manuscript, Mouchabac and colleagues propose a conceptual and critical review of the literature regarding the theoretical and technical principles of digital applications as tools to boost dopamine in individuals with depression. They emphasizing the relevance of using these tools in the treatment of depression. Overall this study has many strengths. The results are interesting and would be of general interest. However, some minor points should be addressed before publication:
- The discussion of the link between dopamine and depression is weak and should be strengthened. In the introduction, Authors might to want to discuss generally the link between dopamine and stress-related disorders such as depression and PTSD (PMID: 29106542; PMID: 31057408), and more specifically the link between dopamine, reward (PMID: 30648615; PMID: 20925949), cognition (PMID: 27256556; PMID: 33167370) and motivation (PMID: 21144997), by adding the references I mentioned.
- As this work should be part of the special issue "Advanced Research on Dopaminergic Neurons and Their Role in Depression" (keywords: dopamine, major depressive disorder, biological substrates of depression, clinical and experimental research), I suggest to add some statements underlining the fact that these digital tools, together with highly translational experimental models (PMID: 31493767; PMID: 33392367), are essential for the treatment of stress-related disorders such as depression and PTSD.
Author Response
we thank the reviewer #1 for his kind comment. Your suggestions will improve the content of the article.
Q1 : The discussion of the link between dopamine and depression is weak and should be strengthened. In the introduction, Authors might to want to discuss generally the link between dopamine and stress-related disorders such as depression and PTSD (PMID: 29106542; PMID: 31057408), and more specifically the link between dopamine, reward (PMID: 30648615; PMID: 20925949), cognition (PMID: 27256556; PMID: 33167370) and motivation (PMID: 21144997), by adding the references I mentioned.
Thank you for this suggestion. We assumed that since this article was part of a special issue on dopamine in depression, this topic would be further developed in other articles. But as you suggested and in order to emphasize the role of dopamine, we have added the following part ( we keeped 4 references) :
"While we cannot reduce each of the clinical dimensions of depression to a single neurotransmitter, it is clear that dopamine deficiency plays a critical role in altered motivated behaviors(5), Belujon et al 2017. Indeed, the dopaminergic system promotes the development of motivation and the triggering of adaptive behaviors(7) . This system is activated by pleasant or aversive stimuli and therefore helps to adapt to changes, thus promoting survival. The attribution of an emotional valence to the events is fundamental and will be supplemented via the reward system(8). Finally, at the cognitive level, dopaminergic neurons are involved in attentional processes and the learning of new situations(9)."
references
Belujon P, Grace AA. Dopamine System Dysregulation in Major Depressive Disorders. Int J Neuropsychopharmacol. 2017 Dec 1;20(12):1036-1046. doi: 10.1093/ijnp/pyx056. PMID: 29106542; PMCID: PMC5716179.
Arias-Carrión O, Stamelou M, Murillo-Rodríguez E, Menéndez-González M, Pöppel E. Dopaminergic reward system: a short integrative review. Int Arch Med. 2010 Oct 6;3:24. doi: 10.1186/1755-7682-3-24. PMID: 20925949; PMCID: PMC2958859.
Torrisi SA, Laudani S, Contarini G, De Luca A, Geraci F, Managò F, Papaleo F, Salomone S, Drago F, Leggio GM. Dopamine, Cognitive Impairments and Second-Generation Antipsychotics: From Mechanistic Advances to More Personalized Treatments. Pharmaceuticals (Basel). 2020 Nov 5;13(11):365. doi: 10.3390/ph13110365. PMID: 33167370; PMCID: PMC7694365.
Bromberg-Martin ES, Matsumoto M, Hikosaka O. Dopamine in motivational control: rewarding, aversive, and alerting. Neuron. 2010 Dec 9;68(5):815-34. doi: 10.1016/j.neuron.2010.11.022. PMID: 21144997; PMCID: PMC3032992.
Q2 : As this work should be part of the special issue "Advanced Research on Dopaminergic Neurons and Their Role in Depression" (keywords: dopamine, major depressive disorder, biological substrates of depression, clinical and experimental research), I suggest to add some statements underlining the fact that these digital tools, together with highly translational experimental models (PMID: 31493767; PMID: 33392367), are essential for the treatment of stress-related disorders such as depression and PTSD.
Thank you for this suggestion ? We added the following sentence in the conclusion (with this ref):
"These digitals tools, together with highly translational experimental models, are essential for the treatment of stress-related disorders such as depression and post-traumatic stress disorder(63)"
Torrisi SA, Lavanco G, Maurel OM, Gulisano W, Laudani S, Geraci F, Grasso M, Barbagallo C, Caraci F, Bucolo C, Ragusa M, Papaleo F, Campolongo P, Puzzo D, Drago F, Salomone S, Leggio GM. A novel arousal-based individual screening reveals susceptibility and resilience to PTSD-like phenotypes in mice. Neurobiol Stress. 2020 Dec 24;14:100286. doi: 10.1016/j.ynstr.2020.100286. PMID: 33392367; PMCID: PMC7772817.
Reviewer 2 Report
Thanks to the editor and to the author for allowing me the possibility to review this very interesting analysis review focusing on digital apps and their involvements on motivation among depressed patients.
This article is very clear and contributive.
I would proposed few comments:
if the introduction is very well written, a description of self determination theory and of the respective role of intrinsic and types of extrinsic motivation in cognitive and social developpement could be of interest.
Furthermore, a description of the overlapp of apathy and depressives symptoms espacially concerning loss of initiative and loss of responsivness should improve the comprehensive mechanisms of potential interest of apps. Besides a description of the cost benefit analysis (wanting/effort) would improve the coprehensive reading.
the methods could be completed by description of inclusion and exclusion criteria for the choosen articles as beacause of the limited articles specifically focusing on apps for depressed patients (for exemple CBT2go?)
concerning the results, a column of comments (by the authors) for table 2 would be interesting to optimize the understanding
the discussion is very interesting, however a discussion of cultural differences and/or age differences for exemple regarding the acceptability could complets it. the impact of digital phenotype and of "digital biomarkers" need to be precised and more developped in depressed patients.
Even if apps to motivates in healthy or non-depressed persons is very interested, a way is to stimulate a competitive purpose (for exemple in sport apps), are there some data to discuss this acceptability and adaptability for depressed patients?
several limitations could be discussed (limited articles, main objective not necessary focusing on depression, heterogeneity of digital phenotypes, long-terms effects and lifestyle changes of risks of relapses)
this article is really innovative and contributive and these comments have set a target to propose way for improvement.
Author Response
Thanks to the editor and to the author for allowing me the possibility to review this very interesting analysis review focusing on digital apps and their involvements on motivation among depressed patients.
This article is very clear and contributive.
We would like to thank reviewer 2 for his reading and his comments which will help improve our manuscript
I would proposed few comments:
Q1 :If the introduction is very well written, a description of self-determination theory and of the respective role of intrinsic and types of extrinsic motivation in cognitive and social development could be of interest.
Thank you for this comment. We agree with the reviewer and we modified the text accordingly by adding the following sentence (in red in the text) in the paragraph 1.2. A Specific Approach is Needed to Treat Conative Disorders. :
"They are more ecological, that is to say not constrained by an external influence and are supported by the theory of self-determination. Motivation in this model depends on the fulfillment of three universal basic needs: autonomy (having the feeling of functioning independently reinforces the commitment of individuals), the competence which corresponds to the feeling of efficiency and control over the environment (obtaining positive feedback from the environment following a task increases motivation) and finally relatedness (strength of the reciprocal link between individuals). The first two needs are strongly linked to the concept of intrinsic motivation, defined by an activity carried out by the subject according to his value system (expected pleasure). It is opposed to extrinsic motivation which corresponds to the pursuit of an activity to achieve an external goal, under the constraint of environmental factors."
Q2- Furthermore, a description of the overlapp of apathy and depressives symptoms espacially concerning loss of initiative and loss of responsivness should improve the comprehensive mechanisms of potential interest of apps. Besides a description of the cost benefit analysis (wanting/effort) would improve the coprehensive reading.
Thank you for this suggestion. We agree with the reviewer and we modified the text accordingly by adding the following sentence (line in red in the text) in the first paragraph (1.1. Motivation Deficit in Depression: a Neglected Dimension)
"Furthermore, we showed an overlap with anhedonia and other symptoms such as apathy, which belongs to motivated behavior disorder and results in a loss of the ability to feel emotions (positive or negative) or initiate spontaneous behaviors. Clinically, there is a loss of interest and pleasure in daily activities living in patients with apathy. Aboulia, very similar to apathy, refers to a lack of will or initiative and can be seen as a disorder of diminished motivation. The condition was originally considered to be a disorder of the will and people suffering from aboulia are unable to act or make decisions independently; however unlike apathy, aboulic patients remain sensitive to the reward if they complete a task. Aboulia can affect decision making, motor behaviors and communication."
"We therefore understand that the motivational dimension is an important target in the treatment of depression and that the specific treatment of conative disorders, including anhedonia, aboulia or apathy is central."
Q3-The methods could be completed by description of inclusion and exclusion criteria for the choosen articles as beacause of the limited articles specifically focusing on apps for depressed patients (for exemple CBT2go?)
We apologize for this lack of precision and we add in the methodology section the inclusion and exclusion criteria:
Inclusions criteria were articles that integrate data collection (EMA) and intervention (EMI) in order to enhance motivation. It could be:
- Both EMA and an EMI component that are not connected to each other.
- Or EMA triggered by another EMA component.
- Or fully or partially automated intervention based on the EMA components allowing tailored interventions in real time.
Exclusion criteria: articles where motivation, mood or activity were not in the evaluated outcomes, or for which no data collection was used (for example, articles with only one intervention and no EMA)
Q4- concerning the results, a column of comments (by the authors) for table 2 would be interesting to optimize the understanding
We thank reviewer 2 for his interesting suggestion, but for ease of reading we suggest not to add a column to the table because it is not possible to orient it horizontally for editing constraints. If we add this column, it will greatly reduce the space and weigh down the reading. In addition, since this is a table presenting the results, it seems complex to add our comments which find a better place in the discussion part.
Q5- the discussion is very interesting, however a discussion of cultural differences and/or age differences for exemple regarding the acceptability could complets it. the impact of digital phenotype and of "digital biomarkers" need to be precised and more developped in depressed patients.
We thank reviewer 2 for his interesting suggestion, as you may have noticed there is few data in the literature, but we followed your suggestion by adding this part:
As reported by Patoz et al.(61) numerous studies suggest that these applications are more appropriated to mild and moderate depression stages, considering that people suffering from severe depression would be unable to use an app. Regarding the acceptability, Litschitz et al.(62) have demonstrated that patients are interested in the app regardless of age and level of education, even if not familial with IT tools.
Q6- Even if apps to motivates in healthy or non-depressed persons is very interested, a way is to stimulate a competitive purpose (for exemple in sport apps), are there some data to discuss this acceptability and adaptability for depressed patients?
We thank the reviewer for his suggestion. In the discussion we approach this problematic, but the lack of data doesn’t allow us to develop this hypothesis. But we added this short sentence :
It has, also, been mentioned the importance of the self-determination theory, beside of gamification, which contributes to the improvement of intrinsic motivation by promoting autonomy, feeling of competence and relationship with other players.
Q7 :several limitations could be discussed (limited articles, main objective not necessary focusing on depression, heterogeneity of digital phenotypes, long-terms effects and lifestyle changes of risks of relapses)
We thank the reviewer for his remark, we had this short part at the end of the discussion, which completes the previous critical aspects
Although some tools have not been designed specifically for depression and do not take into account the different phases of the disease (prodromal symptoms, active phase, relapse, residual symptoms), the field of digital phenotype seems promising. Moreover, it is complicated to draw too hasty conclusions given the small number of randomized trials and the small samples.